# Study of the Effect of Neutral Polysaccharides from Rehmannia glutinosa on Lifespan of *Caenorhabditis elegans*

**DOI:** 10.3390/molecules24244592

**Published:** 2019-12-15

**Authors:** Yanyan Yuan, Nianxin Kang, Qingxia Li, Yali Zhang, Yonggang Liu, Peng Tan

**Affiliations:** College of Chinese Medicine, Beijing University of Chinese Medicine, Beijing 102488, China; 18754310543@163.com (Y.Y.); knx_0423@163.com (N.K.); liqingqia0702@163.com (Q.L.); Ali857@163.com (Y.Z.); Liuyg0228@163.com (Y.L.)

**Keywords:** lifespan extension, insulin/IGF-1 signaling pathway, neutral polysaccharides, *Rehmannia glutinosa*, stress resistance

## Abstract

The problem of an aging society is becoming increasingly acute. Diseases related to aging also come with it. There are some diseases that people can’t treat fundamentally. Therefore, people try to find a natural ingredient from natural medicine to treat these diseases and improve the quality of life of the elderly. With the screening of a large number of traditional Chinese medicines, we found that polysaccharides from *Rehmannia glutinous* (PRG) can prolong the lifespan of *Caenorhabditis elegans* (*C. elegans*). Neutral polysaccharide is the main component of PRG. In the present study, we used a *C. elegans* model to illustrate the stress resistance and lifespan extension effect and mechanism of two kinds of neutral polysaccharide fractions from *Rehmannia glutinosa* (NPRG), respectively called NPRRP and NPRR. Our data showed that two kinds of neutral polysaccharides fractions could extend the lifespan and delay senescence of wild-type worms. Moreover, the mechanism study revealed that NPRG was able to promote the nuclear localization of DAF-16 resulting in the activation of antioxidant enzymatic systems under oxidative stress. We also observed that NPRG didn’t increase the lifespan of mutants with daf-16 portion loss of function, suggesting NPRG prolonging the lifespan partially required the daf-16 gene on the insulin/IGF-1 signaling pathway (IIS). NPRG was found to have no effect on Escherichia coli OP50 (*E. coli* OP50) growth and pharyngeal pump movement of nematodes, indicating that the anti-aging effect of NPRG is not realized by the caloric restriction. However, mRNA levels of daf-2 were remarkably decreased after NPRG treatment. Thus daf-2 lost its inhibitory effect on the expression of daf-16 and had a continuous stimulation effect on the IIS, then prolonged the life of nematodes. Overall, our results illustrated the potential utilization of NPRG as a functional pharmaceutical ingredient to increase stress resistance and extend the life of *C. elegans* via the IIS, which could be developed as a natural supplement agent.

## 1. Introduction

*Rehmannia glutinosa* is a member of the Scrophulariaceae family. It is a traditional Chinese medicinal herb which is widely used in China, Japan, Korea and other Asian countries. Its steamed roots have been used for different medical purposes as traditional Chinese medicine (TCM) for thousands of years. Polysaccharide abundant in *Rehmannia glutinosa* is one important active ingredient and has a wide range of pharmacological activity. Lots of data has shown that it exerts an anti-inflammatory, anti-bacterial, and anti-tumor effect and that it protects cardiovascular function [1]. Accumulating evidence demonstrates that polysaccharides from natural sources, such as fungi, plants and marine flora, have anti-oxidant properties and protect organisms from the damage of free radicals. Acidic polysaccharide of Epimedium polysaccharides improved the survival under increased oxidative stress of nematodes in toxicity by paraquat, against polyQ-mediated neurotoxicity [2]. Astragalus polysaccharides has been shown to alleviate oxidative stress through reducing reactive oxygen species level and malondialdehyde content and increasing superoxide dismutase and glutathione peroxidase activities and also reducing the expression of proapoptotic gene egl-1 in 6-OHDA-intoxicated nematodes and deceasing acetylcholinesterase activity induced by 6-OHDA [3].

The aging process is a progressive deterioration of multiple physiological functions of organisms [4]. *C. elegans* is a model system, widely used to investigate aging and resistance to oxidative stress due to its short life cycle, rapid reproduction rate and clear life-extension mechanism [5]. Due to the fact that the mechanism of life extension of the *C. elegans* model is relatively clear, it is the most studied model organism in the field of aging research. Like many other biological processes, nematode aging is regulated by a number of classical signal transduction pathways, including insulin/igf-1 signal transduction (IIS) [6,7,8], dietary restrictions (CR) [9,10], reduced mitochondrial respiration [11], and signals from the reproductive system [12].

Our previous studies demonstrated that crude polysaccharides isolated from the aqueous extract of *Rehmannia glutinosa* had robust anti-aging and stress resistance effects on *C. elegans* [13]. However, the exact mechanism of its action remains unclear. Therefore, in this study we used two kinds of neutral polysaccharide fractions to further investigate its beneficial effect and action of molecular mechanism. Moreover, the preliminary structure of NPRG was also characterized.

## 2. Results

### 2.1. Identification of NPRG

The carbohydrate and protein contents of two polysaccharides fractions are listed in Table 1. The carbohydrate content for NPRRP and NPRR were 81.22% and 83.35%, respectively. The protein content of NPRRP and NPRR were 19.03% and 11.2%, respectively. These results indicate that these two polysaccharides are composed of a majority of polysaccharides and a small amount of protein.

### 2.2. FR-IR Spectroscopy of NPRG

The infrared spectra of NPRRP and NPRR displayed obvious characteristic peaks of polysaccharides (Figure 1). There was a broadly stretching vibration peak characteristic of hydroxyl groups at 3300–3500 cm^−1^. There was a broadly stretching vibration peak characteristic of –C=O at 1610~1660 cm^−1^. The bands in the region 1150~1075 cm^−1^ indicated a pyranose form of sugars [a]. This region corresponded to ring vibrations overlapped with stretching vibrations of (C–OH) side groups and the (C–O–C) glycosidic band vibration [14,15]. There was no characteristic absorption peak of the keto carbonyl group at ~1730 cm^−1^, indicating that the sample mainly contained neutral polysaccharide. The infrared spectra absorption peaks of the two polysaccharides are almost the same, which indicates that the main chemical groups of the two polysaccharides are basically the same.

### 2.3. Monosaccharide Composition of NPRG

The total ion flow of monosaccharide derivatives of neutral polysaccharides from NPRG under negative ion mode as shown in Figure 2 and Figure 3 and Table 2 showed the monosaccharide composition of NPRRP and NPRR mainly consisted of galactose, glucose, arabinose, mannose and rhamnose. The NPRR was predominantly monosaccharide of galactose and glucose at about 56.85% and 39.51%, with low levels of mannose, arabinose and rhamnose (relative percentages: 0.43%, 0.23% and 0.02%). Similarly, The NPRRP was predominantly monosaccharide of galactose and glucose at about 53.01% and 45.49%, with low levels of mannose, arabinose and rhamnose (relative percentage: 0.60%, 0.46% and 0.15%).

### 2.4. The Effect of NPRG on E. coli OP50 Growth

Previous work showed that the inhibition of *E. coli* OP50, present in these assays as the food source for the nematodes, can lead to the extended lifespan of *C. elegans* [16]. To investigate whether the anti-aging of NPRG is dependent on its anti-microbial effect, we studied the effect of NPRG on *E. coli* OP50 growth. As shown in Figure 4, NPRRP and NPRR had no effect on the growth of *E. coli* OP50 at the doses effective to extend lifespan of *C. elegans*, indicating that the anti-aging effect of NPRG is not realized by the CR generated by inhibiting bacterial reproduction.

### 2.5. Effect of NPRG on Pharyngeal Pump Movement of Nematodes

We used the rate of pharyngeal pump to indicate the food intake, aging and locomotion of nematodes. As shown in Figure 5, compared with the blank group, the pharyngeal pump frequency of the experimental group was slightly increased, but there was no significant difference, indicating that PRG had no effect on the intake of nematodes. Further, the mechanism of PRG prolonging the lifespan of nematodes was not achieved by CR.

### 2.6. Reduction of Age Pigments Accumulation and Extension of Lifespan by NPRG in C. elegans

Age-related auto-fluorescent age pigments, including lipofuscin and advanced glycation end-products, are widely regarded as biomarkers of aging, and their accumulation is shown to be inversely correlated with longevity [17]. As shown in Figure 6a, the spontaneous blue fluorescence of lipofuscin in nematodes was observed under a fluorescence microscope. As shown in Figure 6b, through the calculation of fluorescence density, we found that NPRG could reduce the accumulation of lipobrowcin, which further proved the effect of delaying aging of nematodes.

### 2.7. NPRG Improved the Expression and Activity of Antioxidant Enzymes in Nematode under Oxidative Stress

Studies have shown the role of daf-2/daf-16 in regulating the aging and longevity of *C. elegans*. However, the sod-3 gene that was the downstream factor of the daf-16 gene was regulated by the daf-2/daf-16 gene and was related to lifespan [18]. Therefore, we selected transgenic strain CF1553 that expressed sod-3p::GFP fusion protein. As shown in Figure 7a, after nematodes were exposed to NPRG for 72 h, green fluorescence only appeared in the head and tail of the nematodes. Statistical analysis was carried out on the amount of fluorescence, as shown in Figure 7b, experimental group and blank group, and there was no significant difference between total fluorescence yield, which illustrated that NPRG cannot promote sod-3 gene expression under normal living conditions. Therefore, we studied the effect of NPRG on the expression of the sod-3 gene under paraquat oxidative stress. After being exposed to paraquat for an additional 24 h, nematodes were almost full of green fluorescence under a fluorescence microscope. As shown in Figure 7c, NPRG can significantly increase the total amount of green fluorescent protein compared with the blank group, which indicates that NPRG can increase the expression of the sod-3 gene and enhance the anti-oxidative stress ability under oxidative stress. This further explains the increased survival rate and increased and the decreased ROS content under paraquat oxidative stress in our previous study [13].

Antioxidant enzymes are an important component of antioxidant systems in vivo and play an important role in reducing oxidative damage. For example, SOD can convert superoxide anion to hydrogen peroxide, and CAT can reduce hydrogen peroxide to water. To study the effects of NPRG on the activity in nematodes under oxidative stress, SOD and CAT kits were used to measure the activity of SOD and CAT in nematodes exposed to NPRG under oxidative stress. As shown in Table 3, the experimental results showed NPRG could increase the activity and content of SOD and CAT in nematode under oxidative stress. It is further proved that under the condition of oxidative stress, NPRG can stimulate the expression of sod-3 and other antioxidant related genes, enhance those gene’s transcription levels to produce more antioxidant enzymes, remove excessive ROS and improve the level of antioxidant stress in nematodes.

### 2.8. NPRG Can Induce the Nuclear Localization of DAF-16::GFP under the State of Oxidative Stress

The mammalian DAF-16 orthologs regulate genes involved in growth control, apoptosis, DNA repair, and oxidative stress [19]. Combined with our previous results, we predicted that IIS was responsible for NPR’s effects of stress-resistance and life-prolonging on *C. elegans*. By using transgenic worms expressing daf-16a/b::GFP fusion protein, the regulation of daf-16 transcriptional activity by NPRG was studied. Under normal growth conditions, DAF-16 is mainly retained in the cytoplasm. NPRG did not promote the nuclear localization of DAF-16 but distinctly promoted nuclear localization under the oxidative stress state of paraquat, as shown in Figure 8a. By counting fluorescence points, NPRR and NPRRP did increase the expression of daf-16 in the nucleus compared to the blank group as shown in the Figure 8b. Under normal living conditions, NRP cannot directly activate daf-16. Once the nematodes were under stress, NPR could increase the expression of DAF-16, significantly.

### 2.9. NPRG Inhibited daf-2 Gene Expression in the IIS

The aging process of *C. elegans* is regulated by several signaling pathways [20]. Among all the pathways regulating *C. elegans* aging, IIS might be the most well-studied. Daf-2, a key component of IIS, has been found to extend the lifespan of *C. elegans*. Transcription factor Daf-16 extends lifespan through regulation of longevity genes [21,22]. In the IIS, daf-2 can negatively regulate this pathway. When the transcriptional activity of daf-2 is inhibited, the nucleation expression of daf-16 downstream of daf-2 will be increased, to achieve the effect of anti-pressure and lifespan extension [23,24]. RT-PCR was used to detect the effects of NPRG on the expression levels of daf-2, tub-1, daf-16, sod-3, ctl-1 and hsp-16.2 mRNA. As shown in Figure 9, daf-2 expression in nematodes decreased after PRG administration for 3 days compared with the blank group, and the expression levels of tub-1, daf-16, sod-3, ctl-1 and hsp-16.2 remained basically unchanged. This suggests that the effect of NPRG in delaying senescence may be by inhibiting the transcriptional activity of daf-2. Under normal conditions, NPRG extends the lifespan of *C. elegans* by inhibiting the expression of the daf-2 gene in the IIS.

### 2.10. NPRG Can Extend the Lifespan of daf-16 Mutant DR26, but Cannot Increase Survival Rate of daf-16 Mutant CF1038

To investigate whether the life-prolonging effect of NPRG depends on the existence of daf-16, a long-lived gene, we selected daf-16 mutant strains DR26 and CF1038 with modified daf-16 gene for the longevity experiment. According to the results of the survival curve in Figure 10a,b, under the continuous exposure of NPRG, the maximum longevity of DR26 was about 22 d, which was similar to N2, but the maximum longevity of CF1038 did not increase. This indicates that the anti-aging effect of NPRG is dependent on IIS, and it is directly related to part of the daf-16 gene fragment.

## 3. Discussion

To investigate the lifespan extension mechanism of NPRG against nematodes, we eliminated the caloric restriction pathway by investigating the effects of polysaccharides on the growth of *E. coli* and the food intake of *C. elegans*. Through the determination of expression and activity of antioxidant enzymes, we found that NPRG can improve sod-3 gene expression and increase the activity of antioxidant enzymes under oxidative stress state. The increase of antioxidant enzyme activity is the main reason for the survival rate of nematodes under oxidative stress. When nematodes are subjected to external heat and oxidative stress, daf-16 will rapidly transfer to the nucleus under the stimulation of these stress signals and then induce the high expression of these downstream effector elements so as to resist the pressure caused by external stress and thus extend the lifespan of nematodes [25,26,27,28]. Further experiments showed that under the normal condition of nematodes, NPRG did not induce nuclear translocation of daf-16::GFP, but under the condition of oxidative stress, NPRG promoted nuclear translocation of daf-16::GFP, which indicates that the anti-aging reason of PRG might be to extend the lifespan of nematodes by regulating the daf-16 gene. We confirm the conclusion that daf-16 mutants could not be prolonged by daf-16 mutants through the lifespan assay of daf-16 mutants. It’s worth noting that NPRG’s effect of prolonging the lifespan of nematodes is directly related to part of the daf-16 gene fragment. DAF-16 is known to have several subtypes: DAf-16f (r13h8.1f), DAf-16d (r13h8.1d), DAF-16a1(r13h8.1b), DAF-16a2 (r13h8.1c), DAF-16b (r13h8.1a). DAF-16 transcription products daf-16a, daf-16b and daf-16d/F are all involved in dauer formation, fat storage and anti-stress. However, for life regulation, only daf-16a and daf-16d/f was found to be effective [29,30,31]. Based on the location of the m26 gene, we speculated that only daf-16a was responsible for the effect of prolonging the lifespan of nematodes with NPRG treatment.

Through the RT-PCR experiment, we found that PRG could significantly inhibit the transcriptional activity of daf-2 but had little effect on other genes, which indicates that NPRG could inhibit the negative regulation of the insulin signal receptor daf-2 on the IIS in nematodes by inhibiting the expression of daf-2, thereby extending the lifespan of nematodes. A lot of literature has shown that Daf-2 is located on the cell membrane as an insulin receptor [32,33,34,35]. When activated, it activates phosphatidylinositol-3 kinase homolog age-1, which causes downstream AKT to be activated by the second messenger produced by age-1, thus phosphorylating the transcription factor daf-16. Phosphorylated daf-16 cannot enter the nucleus to play a transcriptional regulatory role [24,36]. Therefore, when daf-2 is knocked out, its negative regulation effect on daf-16 is removed, and daf-16 enters the nucleus to play a transcriptional regulation function, thus extending the lifespan of nematodes [25,37].

In conclusion, NPRG prolonged the life of nematodes by IIS and acted on daf-2 and daf-16 for key genes on this pathway. NPRG continuously inhibits the transcriptional activity of daf-2, thus continuously inhibits the activity of this pathway, which achieves the function of extending life span, under normal living conditions. Once the worms were under stress, NPRG enhanced the transcriptional activity of daf-16 to activate the expression of downstream target genes to play an antihypertensive role.

## 4. Materials and Methods

### 4.1. C. elegans Culture Conditions

The strains used in this study were wild-type N2; CF1308, daf-16(mu86); TJ356, zIs356 [daf-16p::daf-16a/ b::GFP+ rol-6(su1006)]; CF1553, muIs84[(pAD76) sod-3p::GFP +rol-6(su1006)]; DR26, daf-16(m26) (obtained from the Chinese Academy of Sciences). All strains were maintained on nematode growth medium (NGM) plates seeded with *E. coli* OP50 at 20 °C as described [38]. The nematodes were transferred to a fresh NGM plate every 2 days. Age-synchronized *C. elegans* were obtained through the sodium hypochlorite method [39] and gravid adults lay eggs method.

### 4.2. Chemicals and Reagents

Diethyl aminoethyl cellulose-52 (DEAE-52) was purchased from Solarbio Co. (Beijing, China), batch number 20181109; Sephadex G-100 (batch number 17-0060-02) was purchased from Biotopped Co. (Beijing, China). Assay kits for SOD (batch number 20190319), BCA (batch number 20190411) were purchased from Nanjing JianCheng Bioengineering Institute (Nanjing, China). CAT (batch number 20190429) was purchased from Sigma Chemical Co. (St. Louis, MO, USA). All other reagents are all analytical grade.

### 4.3. Preparation of NPRG

In our previous experiments, the preparation method of polysaccharides from Rehmanniae radix preparata (PRRP) and Rehmanniae radix (PRR) was recorded [13]. RR and RRP was cut into 0.5 cm^2^ small pieces, and 10 times more acetone was added to purge fat-soluble components. We added 10 times more water to pigment and decocted it on a radiant cooker 3 times, each time for 1 h. Then, we merged the decoctions and concentrated the filtrate using reduced pressure distillation at 70 °C to an appropriate amount. Next, we added 95% ethanol until the content of it in filtrate was 80%, placing it for 12 h under 4 °C. Then, we filtered and froze it to dry. We dissolved it in distilled water and deproteinized it using sevage (chloroform: n-butanol = 4:1) and repeated this more than 10 times until no protein absorption was detected using UV spectrum analysis. Finally, the PRG were obtained by freeze-drying. 

NPRRP and NPRR polysaccharide fractions were previously isolated from *Rehmannia glutinosa* crude polysaccharides in our lab. Briefly, PRRP and PRR were dissolved in distilled water and then loaded onto a DEAE-52 cellulose column (2.0 × 20 cm) to stepwise elute with deionized water with a flow rate of 0.6mL/min. The elution was concentrated, dialyzed, lyophilized, and further purified on a Sephadex G-100 column (3.5 × 25 cm). The gel permeation chromatography (GPC) analysis showed that NPRRP and NPRR respectively appeared as a single and symmetrical peak, indicating that NPRRP and NPRR were comparatively homogeneous polysaccharides 

### 4.4. Chemical Composition Analysis

The carbohydrate content was measured using the phenol-sulfuric acid method with D-glucose as the standard [40]. The protein content was detected according to the instructions on the kit.

### 4.5. FT-IR Spectroscopy

The FT-IR spectra of NPRRP and NPRR were recorded using a Fourier infrared spectrometer (Nicolet is10, Thermo, USA) with the frequency range of 4000–400 cm^−1^. The polysaccharides were mixed with spectroscopic grade potassium bromide powders and then were pressed into 1.0 mm pellets [41].

### 4.6. Analysis of Monosaccharide Compositions

The monosaccharide compositions of NPRRP and NPRR were carried out as previously described with minor modifications [42]. The UPLC analysis was carried out on AcquityTM ultra high-performance liquid chromatogram system (Waters, USA). The separation was carried out on an analytical C18 column (Acquity BEH C18, 2.1 mm × 100 mm, 1.7 um). The mobile phase consisted of ammonium acetate of 20 mmol/L whereby 1 m L of glacial acetic acid was added for every 100 mL (A) and acetonitrile (B) using an isocratic elution of 85% A and 15% B. The column temperature was maintained at 30 °C. The temperature of the sample room was 4 °C. The flow rate was 0.2 mL/min. Detection wavelength was 254 nm. Injection volume was 2ul. Mass spectrometry analysis was performed using a SYNAPT G2-SI QTOF-MS (Waters, USA) equipped with electrospray ionization (ESI) source. The mass spectrometry parameters were as follows: ion source temperature, 100 °C; temperature of solvent removal, 400 °C; flow rate of dissolvent gas, 600 L/h; conical gas flow rate, 60 L/h; taper hole voltage, 40 V; capillary voltage 3.0 KV high energy channel; collision voltage, 10–60 Ev. Negative ion scanning mode was adopted. Mass sweep range was m/z 50‒1200. Scanning time was 0.2 s. Mass Lynx V4.1 software was used for data collection and analysis.

### 4.7. Exposure Experiments

The polysaccharide was added to the *E. coli* OP50 fluid at a concentration of 5 mg/ml. The fluid was added to the surface of NGM. Nematodes were inoculated in the synchronization process in the egg or L4 period and then cultured under 20 °C. In the control group, nematodes were cultured on the NGM with *E. coli* OP50 containing distilled water added.

### 4.8. Bacterial Growth Assay

*E. coli* OP50 was seeded into sterilized liquid medium with or without 50 μg/ml BSP, and initial OD595 values were measured against sterile medium. Samples were transferred to a rocking shaker at 37 °C, and OD595 values were measured once every 12 h up to 72 h.

### 4.9. Pharyngeal Pump Frequency Assay

The synchronized nematodes were continuously exposed to the polysaccharides, 20 nematodes were randomly selected from each group on day 4 and day 8 of lifespan, and the pharyngeal pump movement times were measured for 20 s.

### 4.10. Lipofuscin Assay

After exposing to polysaccharides for 10 d, 20 nematodes in each group were randomly selected and photographed after purple fluorescence excitation with a fluorescence microscope (ECLIPSE, TSR, Nikon) at excitation/emission wavelengths of 380 and 430 nm. The content of lipofuscin was measured using ImageJ (National Institutes of Health, Bethesda, MD, USA).

### 4.11. Visualization of SOD-3::GFP

The transgenic strain CF1553 worms were employed to measure expression of sod-3. Synchronized worms were maintained with NPRRP and NPRR from eggs. After 72 h of exposure to NPRRP and NPRR, the GFP fluorescence of worms was examined using fluorescence microscopy with excitation/emission wavelengths of 485 and 530 nm. The fluorescence intensity was quantified using ImageJ software. A minimum of 20 worms per group was used in each experiment. Assays were performed in independent triplicates. To further study the effect of NPRG on the expression of sod-3 gene under oxidative stress, nematodes were exposed to paraquat containing 70 uM for another 24 h.

### 4.12. Measurement of Antioxidant Enzyme Activities

Determination of SOD and catalase CAT content and activity were performed as previously described [43]. Approximately 2000 eggs were incubated with or without polysaccharide and 70 mM paraquat as above. The nematodes were collected and homogenized in M9 solution. The lysate was collected and used for protein quantification using a BCA kit as above, for SOD activity assay using Total Superoxide Dismutase Assay Kit (Jiancheng, Nanjing, China), and for CAT activity assay using Total Superoxide Dismutase Assay Kit (St. Louis, MO, USA). The SOD and CAT activity were normalized by protein content.

### 4.13. Nuclear Localization of DAF-16

TJ356 strain carrying a daf-16::GFP was used to test the translocation of DAF-16 in the nucleus. Synchronized worms of L4 were maintained on NGM plates with or without NPRG for 24 h. TJ356 worms were fixed on a microscopy slide with 20 mM sodium azide and subcellular DAF-16 distribution was observed under the fluorescence microscope. Intracellular location of DAF-16 was categorized as cytosolic, intermediate (‘both nuclear and cytosolic’) and nuclear. A minimum of 20 worms per group was used in each experiment. To further study the effect of NPRG on the expression of the daf-16 gene under oxidative stress, nematodes were exposed to paraquat containing 70 uM for another 24 h. The number of fluorescence points in each nematode was expressed as the expression of DAF-16 in the nucleus.

### 4.14. RT-PCR Assay

Approximately 2000 eggs were incubated with or without polysaccharide for 72 h. Total TRNzol RNA extraction reagent was used for sample RNA extraction. RNA was tested by ultraviolet absorption assay and denature-agar gel electrophoresis. Then a PrimeScript™ RT reagent kit with gDNA eraser was used for cDNA reverse transcription. RT-PCR was used to determine the relative quantification of the targeted genes in comparison to the reference act-1 gene, and the results were expressed as the relative expression ratio (between targeted gene and internal control act-1). Primer sequences for the genes of interest were as follows:

daf-2, F 5’-CCAACCGAACGGAGACCT-3’, R 5’-CGATAGCCGAACACGAAC-3’;

daf-16, F 5’-CGTTTCCTTCGGATTTCA-3’, R 5’-ATTCCTTCCTGGCTTTGC-3’;

tub-1, F 5’-AGTGCGGGAAGCGTGAGA-3’, R 5’-GCATCGACTGCATACGTGGT-3’;

sod-3, F 5’-ACTTGGCTAAGGATGGTGGAG-3’, R 5’-CCTTGAACCGCAATAGTGATG-3’; hsp-16.2, F 5’-CGCTATCAATCCAAGGAGAAC -3’, R 5’-GAAGCAACTGCACCAACATC-3’; ctl-1, F 5’-TCCTACACGGACACGCATTAC-3’, R 5’-CGGAAACTGTTCGGGAAGTAA-3’; act-1, F 5’-TGACGGACAAGTCATCACCG-3’, R 5’-CATGGTGGTTCCTCCGGAAA-3’.

### 4.15. Lifespan Assay of daf-16 Mutant of Nematodes

For lifespan, 30 nematodes of L4 were randomly transferred to fresh NGM plates treated with the NGM at 20 °C. The day of synchronization was day 2 in the survival curve. Nematodes were transferred to fresh plates every two days and scored as alive, dead or lost every day. Death was defined as no responsiveness to gentle mechanical touch. 

### 4.16. Statistical Analyses

All the tests were carried out in parallel three times and the average value of the three times was taken for the result analysis. SAS 8.2 statistical software SAS 8.2 (SAA institute, Gary, NC, USA) was used to analyze the difference between the groups using a t-test, and differences between groups were determined using analysis of variance. GraphPad Prism 5 (Prism, GraphPad Software, San Diego, CA) was used for the survival curves and survival analysis. *P* < 0.05 was considered statistically significant.

## 5. Conclusions

Our study has shown that NPRG can resist pressure and prolong life. The mechanism of action of NPRG to extend the lifespan of *C. elegans* may be relatively clear. On the one hand, NPRG continuously inhibits daf-2 transcriptional activity, thereby inhibiting IIS, which daf-2 negatively regulates, under normal conditions. On the other hand, NPRG can stimulate the transcription factor daf-16 more into the nuclear expression, to prolong the life and enhance the anti-stress ability of nematodes via the expression of downstream target genes that have a corresponding function, under the state of stress. In addition, it is worth further investigating whether daf-16a was responsible for the effect of prolonging the lifespan of nematodes with NPRG treatment. Taking these data together, NPRG exhibited a beneficial effect on the aging process and could be further studied as a promising natural antioxidant.

## Figures and Tables

**Figure 1 molecules-24-04592-f001:**
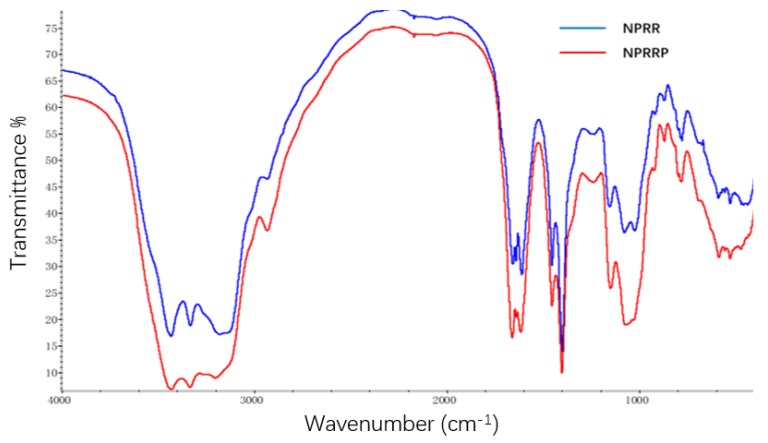
Infrared spectra of polysaccharides: NPRR, NPRRP.

**Figure 2 molecules-24-04592-f002:**
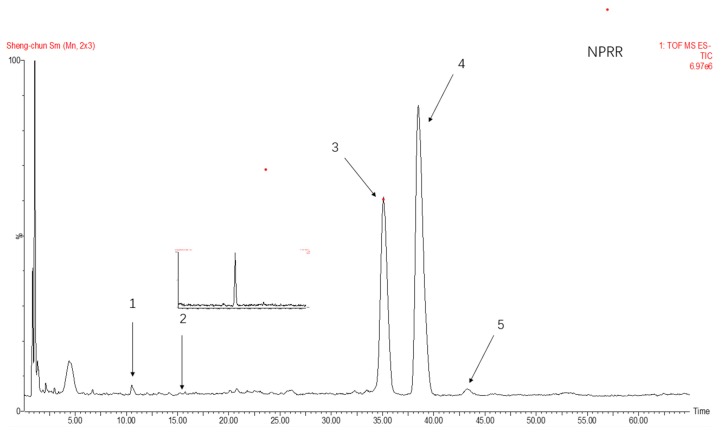
The total ion flow of monosaccharide derivatives of neutral polysaccharides from NPRR under negative ion mode. Number 1–5 respectively refers to mannose, rhamnose, glucose, galactose, arabinose.

**Figure 3 molecules-24-04592-f003:**
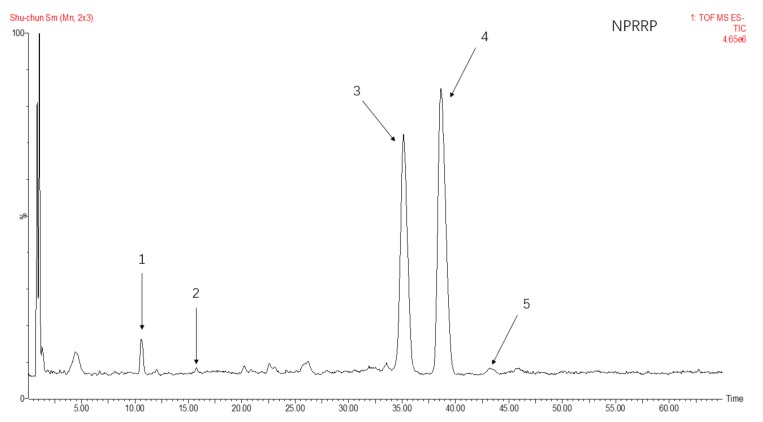
The total ion flow of monosaccharide derivatives of neutral polysaccharides from NPRRP under negative ion mode. Number 1–5 respectively refers to mannose, rhamnose, glucose, galactose, arabinose.

**Figure 4 molecules-24-04592-f004:**
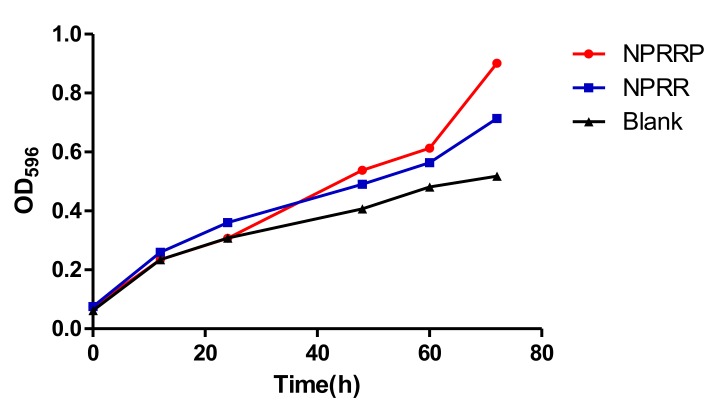
The effect of NPRG on *E. coli* OP50growth. Escherichia coli OP50 was seeded to medium with or without NPRG and then transferred to rocking shaker at 37 °C. OD595 values were measured for 72 h and growth curves of Escherichia coli OP50 in both mediums were prepared. The data represents three parallel experiments.

**Figure 5 molecules-24-04592-f005:**
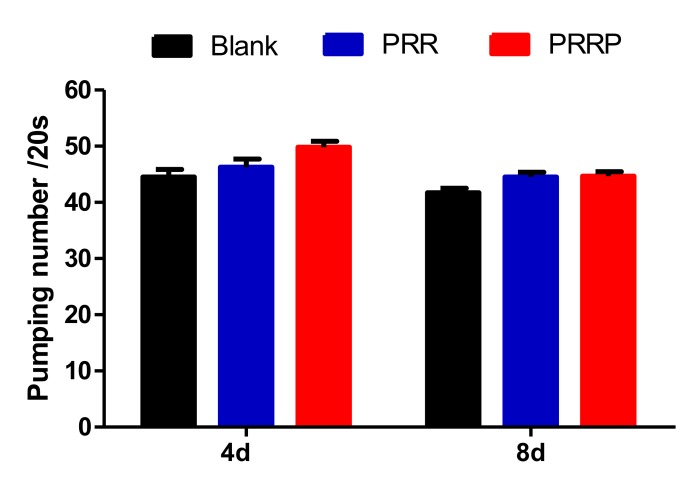
The effect of PRG on the pharyngeal pumping rate of *C. elegans*. The pharyngeal pump times of nematodes in 20 s were measured on fourth and eighth days of the nematodes’ life respectively to indicate the effects of NPRG on nematodes’ feeding, aging and locomotion. Compared with the blank group, the pharyngeal pump frequency of the experimental group was slightly increased, but there was no significant difference. The data represents three parallel experiments.

**Figure 6 molecules-24-04592-f006:**
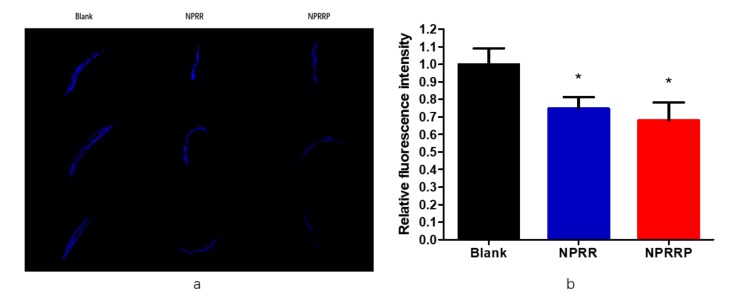
(**a**) The blue fluorescence produced by lipofuscin in nematodes under fluorescent irradiation. (**b**) Effects of NPR on lipofuscin in nematodes. To further the anti-aging effect of NRP on nematodes, we used image J to measure the blue fluorescence intensity to indicate the content of lipofuscin in nematodes. * *p* < 0.05, NPRR vs. Blank; * *p* < 0.05, NPRRP vs. Blank.

**Figure 7 molecules-24-04592-f007:**
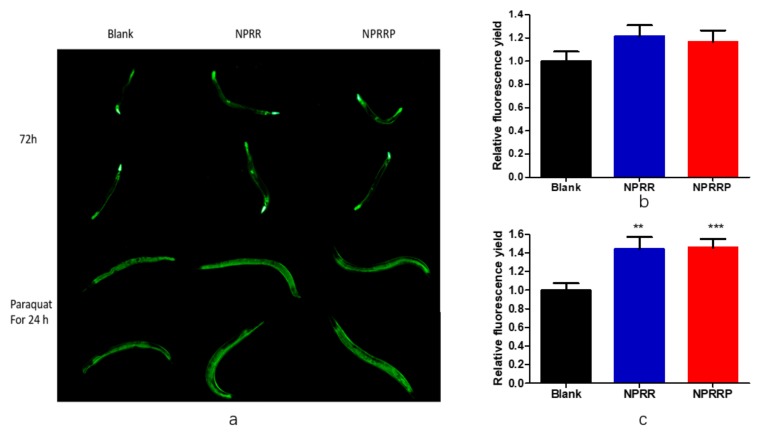
(**a**) The green fluorescence produced by CF1553 nematode under fluorescent irradiation. (**b**) Effects of polysaccharides on fluorescence expression of CF1553 nematodes under normal living conditions. Image J was used to measure the relative fluorescence yield to indicate the expression of sod-3 in nematodes. There was no significant difference between the experimental group and the blank group. (**c**) Effects of polysaccharides on fluorescence expression of CF1553 nematodes under oxidative stress. Image J was used to measure the relative fluorescence yield to indicate the expression of sod-3 in nematodes. ** *p* < 0.01, NPRR vs. Blank; *** *p* < 0.001, NPRRP vs. Blank.

**Figure 8 molecules-24-04592-f008:**
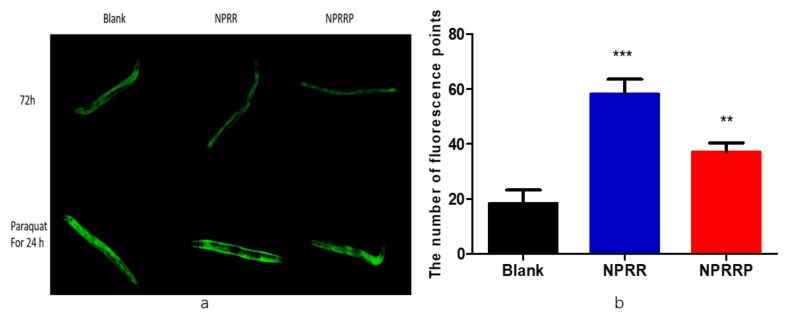
(**a**) The localization of DAF-16::GFP under fluorescent irradiation. Under normal growth conditions, DAF-16 is mainly retained in the cytoplasm. When daf-16 was expressed in the nucleus, the fluorescence signal of DAF-16::GFP was gathered in the nucleus, and the whole nematode was observed under the microscope with a spot-like green fluorescence. (**b**) The number of fluorescence points of DAF-16::GFP under the oxidative stress state of paraquat. The number of fluorescence points represented the amount of expression of daf-16 into the nucleus. *** *p* < 0.001, NPRR vs. Blank; ** *p* < 0.01, NPRRP vs. Blank.

**Figure 9 molecules-24-04592-f009:**
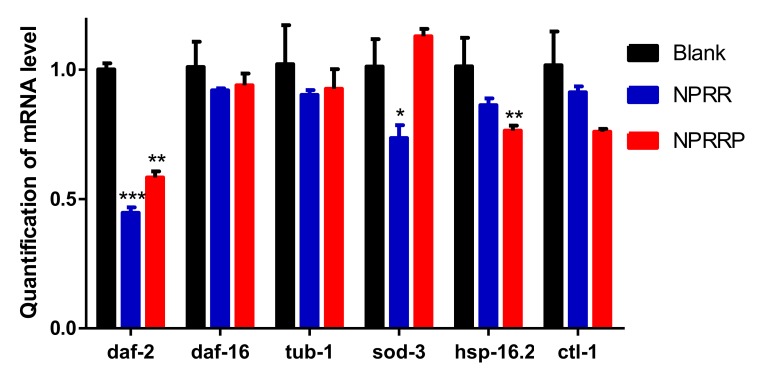
Influence of NPRR and NPRRP on the mRNA expression of daf-2, tub-1, daf-16, sod-3, hsp-16.2 and ctl-1 in *C. elegans* N2.

**Figure 10 molecules-24-04592-f010:**
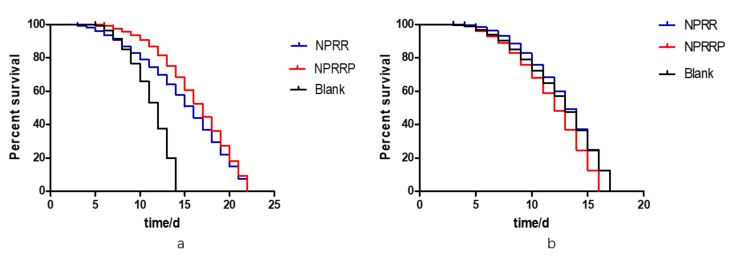
(**a**) Survival curves of DR26 nematodes exposed to NPRG. (**b**) Survival curves of CF1038 nematodes exposed to NPRG.

**Table 1 molecules-24-04592-t001:** The contents of carbohydrate and protein of NPRRP and NPRR.

Sample	NPRRP	NPRR
Carbohydrate (%)	81.22	83.55
Protein (%)	19.03	11.20

**Table 2 molecules-24-04592-t002:** Monosaccharide composition and relative percentage (%) of NPRR and NPRRP.

	Mannose	Rhamnose	Glucose	Galactose	Arabinose
NPRR	2.38	0.05	39.51	56.85	1.22
NPRRP	0.43	0.02	44.08	55.23	0.23

**Table 3 molecules-24-04592-t003:** Effects of NPRG on antioxidant enzyme activity of *C. elegans.*

Sample	SOD (U/mg)	CAT (U/mg)
Blank	1.83 ± 0.035	5.68 ± 0.518
NPRR	3.54 ± 0.275	10.67 ± 1.848
NPRRP	2.66 ± 0.096	11.27 ± 3.037

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
