# Peer review of "Study of the Effect of Neutral Polysaccharides from Rehmannia glutinosa on Lifespan of Caenorhabditis elegans"

_molecules, 2019, doi:10.3390/molecules24244592_

Round 1
Reviewer 1 Report
The authors have done a systematic pharmacological study for two neutral polysaccharide fractions from Rehmannia glutinosa on stress resistance and lifespan extending by applying C. elegans model. I have some comments for the authors’ consideration regarding the figure, experimental design, data quality and organization of manuscript.
In the Introduction part, the authors need to provide more background for the application of elegans model in aging research. There are lots of typos throughout the manuscript, I suggest the authors to go for careful proof reading before their manuscript submission in future.
Typically,
Line 152, Page 7: what do you mean by MDA?
Line 202, Page 9: what does Figure 14 refer to?
Line 197, Page 9; Line 105, Page 4: C. elegans should be written in italic.
Line 192, Page 9: what do you mean by oleanolic acid?
Figure 11: there is no P-value or error bar for each group especially for daf-2 group. Do you mean none of the comparison is statistically significant? Figure 2: have you done replicate experiments for each time-point? Result 2.10 on Page 9, what is the relationship between daf-16 mutant TJ356, CF1038 and DR26? The authors need to discuss more about this in discussion part. Result 2.9 on Page 8: have you done the experiment to test whether NPGR can inhibitdaf-2 expression under oxidative stress? Please check the number of Figure 12 and 13 In the Method part 4.3 on Page 11: the authors need to provide detailed preparation process of NPRRP and NPRR. The structure of the two NPRGs were not fully characterized as the information of molecular weight, sugar chain structure and linkage form are missing. The authors can refer to the literature below. Result 2.3 on Page 3: The authors need to show out the LC chromatograms of NPRR and NPRP for monosaccharide composition analysis. There are too many figures included in this manuscript. I suggest to combine Figure 2-5, Figure 6-8 and Figure 9-13 according to the content described in each.
https://www.nature.com/articles/s41598-018-38251-6
Author Response
Thank you for your valuable comments on my paper and pointing out the mistakes and shortcomings.
Reviewer Number |
Original comments of the reviewer |
Reply by the author(s) |
1 |
Line 152, Page 7: what do you mean by MDA? |
Changed it to CAT. |
2 |
Line 202, Page 9: what does Figure 14 refer to? Please check the number of Figure 12 and 13 . |
Changed it to Figure 13, I have checked the number of Figure 12 and 13 to ensure correct. |
3 |
Line 197, Page 9; Line 105, Page 4: C. elegans should be written in italic. |
Correction has been made. |
4 |
Line 192, Page 9: what do you mean by oleanolic acid? |
Changed it to NPRG. |
5 |
Figure 11: there is no P-value or error bar for each group especially for daf-2 group. Do you mean none of the comparison is statistically significant? |
Sorry, in the beginning, I plotted with average, but after your reminding, I realized that it was not rigorous, so I plotted with the original data and adopted the analysis of variance (ANOVA) by SAS 8.2. |
6 |
Figure 2: have you done replicate experiments for each time-point? |
Yes, I have done replicate experiments for three times for each time-point, and plotted with average. |
7 |
Result 2.10 on Page 9, what is the relationship between daf-16 mutant TJ356, CF1038 and DR26? The authors need to discuss more about this in discussion part. |
I am so sorry, in result 2.10 there shouldn't have been TJ356.I have changed it to CF1038. In addition, We have explained to daf-16 mutant TJ356 in 4.13. |
8 |
Result 2.9 on Page 8: have you done the experiment to test whether NPGR can inhibit daf-2 expression under oxidative stress? |
We didn't do it. Our aim was to investigate the mechanism by which NPRG prolongs the lifespan of nematodes under normal conditions. But testing whether NPGR can inhibit daf-2 expression under oxidative stress can further enrich the research results. We will study it in the following experiment. |
9 |
In the Method part 4.3 on Page 11: the authors need to provide detailed preparation process of NPRRP and NPRR. |
We added the following: RR and RRP will be cut into 0.5 cm2 small pieces, and 10 times more acetone was added to purge fat‑soluble components. We added 10 times more water to pigment and decoct it on a radiant cooker for 3 times, each time for 1 h. Then, we merged the decoctions and concentrated the filtrate by reduced pressure distillation at 70°C to an appropriate amount. Next, we added 95% ethanol until the content of it in filtrate as 80%, placing it for 12 h under 4°C. Then, we filtered and froze it to dry. We dissolved it in distilled water and deproteinized by Sevage (chloroform: n‑butanol = 4:1) and repeated more than 10 times until no protein absorption was detected by UV spectrum analysis. Finally, the PRG were obtained by freeze‑drying. |
10 |
The structure of the two NPRGs were not fully characterized as the information of molecular weight, sugar chain structure and linkage form are missing. The authors can refer to the literature below. |
Thanks very much for your valuable comment. Molecular weight, sugar chain structure and linkage form of NPRG are important. However, the paper mainly focuses on pharmacological studies, supplemented by chemical structure studies. We will study it in the following experiment. |
11 |
Result 2.3 on Page 3: The authors need to show out the LC chromatograms of NPRR and NPRP for monosaccharide composition analysis. |
I added the figure 2 and figure 3 to show out the LC chromatograms of NPRR and NPRP for monosaccharide composition analysis in result 2.3. |
12 |
There are too many figures included in this manuscript. I suggest to combine Figure 2-5, Figure 6-8 and Figure 9-13 according to the content described in each. |
I combined Figure 6-7, Figure 8-10, Figure 11-12 and Figure 14-15 according to the content described in each. |
Thanks again, best wishes to you!
Reviewer 2 Report
In this manuscript, authors demonstrated that two kinds of neutral polysaccharide fraction form Rehmannia glutinosa (NPRG), NPRRP and NPRR, enhanced tolerance against oxidative stress and prolonged survival through nuclear localization of daf-16 in Caenorhabditis elegans (C. elegans). The subject of study seems to be interesting. However, there are some concerns in this study. The reviewer’s comments and questions are described as follows.
1. Authors suggested “anti-aging” effects of NPRG in C. elegans. However, they just showed beneficial effects of NPRG on oxidative stress and survival. Survival prolongation is not equal to anti-aging. If authors would like to suggest “anti-aging effect”, they have to examine other aging or senescence-associated markers such as p21 and senescence-associated β-galactosidase. Or, they need to remove the term “anti-aging effect”.
2. Authors demonstrated that NPRG increased total superoxide dismutase (SOD) activity. However, they did not show that the increased SOD activity was mainly ascribed for upregulation of SOD3 but not SOD2. Authors have to examine the expression of not only SOD3 but also SOD1 and SOD2, and clearly explain why they focused on SOD3 expression in this study.
3. Quantification of GFP fluorescence seems to be arbitrary. Nuclear localization of DAF-16 should be confirmed by other methods such as western blot after the separation of nuclear fraction from cytoplasmic fraction. In addition, as nuclear localization of DAF-16 is regulated by phosphorylation process, western blot for phospho-DAF-16 relative to total DAF-16 should be helpful.
4. In Figure 11, why were standard deviations not shown? How many samples were examined?
5. In statistical analysis, t-test is inappropriate for comparing the differences among multiple groups. ANOVA with post hoc tests should be used. Authors should retry their statistical analyses in this study.
Author Response
Thank you for your valuable comments on my paper and pointing out the mistakes and shortcomings.
Reviewer Number |
Original comments of the reviewer |
Reply by the author(s) |
1 |
Authors suggested “anti-aging” effects of NPRG in C. elegans. However, they just showed beneficial effects of NPRG on oxidative stress and survival. Survival prolongation is not equal to anti-aging. If authors would like to suggest “anti-aging effect”, they have to examine other aging or senescence-associated markers such as p21 and senescence-associated β-galactosidase. Or, they need to remove the term “anti-aging effect”. |
We adopted your valuable opinions and suggestions. And some modifications have been made in the title, abstract, keywords discussion. |
2 |
Authors demonstrated that NPRG increased total superoxide dismutase (SOD) activity. However, they did not show that the increased SOD activity was mainly ascribed for upregulation of SOD3 but not SOD2. Authors have to examine the expression of not only SOD3 but also SOD1 and SOD2, and clearly explain why they focused on SOD3 expression in this study. |
Coleen t. Murphy et al. analyzed the expression of downstream genes of C. elegans daf-16 using DNA microarray analysis and RNAi technology, and proposed the role of daf-2/daf-16 in regulating the aging and longevity of C. elegans, and proved that sod-3 gene was the downstream factor of daf-16 gene [1].YOKO HONDA's study showed that the expression of sod-3 gene was significantly up-regulated in daf-2 mutant nematode, but the expression of sod-1 and sod-2 gene was not significantly changed. Therefore, it could be inferred that the sod-3 gene was regulated by daf-2/daf-16 gene, and was related to the life span of C. elegans[2]. 1. Murphy, C. T.; McCarroll, S. A.; Bargmann, C. I.; Fraser, A.; Kamath, R. S.; Ahringer, J.; Li, H.; Kenyon, C., Genes that act downstream of DAF-16 to influence the lifespan of Caenorhabditis elegans. 424, (6946), 277-283. 2. Honda Y , Honda S . The daf-2 gene network for longevity regulates oxidative stress resistance and Mn-superoxide dismutase gene expression in Caenorhabditis elegans.[J]. The FASEB journal : official publication of the Federation of American Societies for Experimental Biology, 1999, 13(11):1385-1393. Because part of our research results showed that NPRG acted through the IIS signaling pathway, we only focused on examine SOD-3 expression. This is similar to the experimental method in the following paper. 3. Jia, W.; Guangzhi, Z.; Xiaobing, H.; Zhe, W.; Ninghua, T., 1,4-Naphthoquinone Triggers Nematode Lethality by Inducing Oxidative Stress and Activating Insulin/IGF Signaling Pathway in Caenorhabditis elegans. Molecules 22, (5), 798-. 4. Lee, E. B.; Xing, M. M.; Kim, D. K., Lifespan-extending and stress resistance properties of brazilin from Caesalpinia sappan in Caenorhabditis elegans. Archives of Pharmacal Research. 5. Feng, S.; Cheng, H.; Xu, Z.; Yuan, M.; Huang, Y.; Liao, J.; Yang, R.; Zhou, L.; Ding, C., Panax notoginseng polysaccharide increases stress resistance and extends lifespan in Caenorhabditis elegans. Journal of Functional Foods 45, 15-23. |
3 |
Quantification of GFP fluorescence seems to be arbitrary. Nuclear localization of DAF-16 should be confirmed by other methods such as western blot after the separation of nuclear fraction from cytoplasmic fraction. In addition, as nuclear localization of DAF-16 is regulated by phosphorylation process, western blot for phospho-DAF-16 relative to total DAF-16 should be helpful. |
When DAF-16::GFP enters the nucleus, it produces a bright spot of light under a fluorescence microscope. We measured the amount of fluorescence points in each group at the same time to indicate the expression of DAF-16. We think this is a convenient operation to reduce the fluorescence error of software measurement. Thanks very much for your valuable suggestions. TJ356 strain carrying a daf-16::GFP was used to test the translocation of DAF-16 in the nucleus. The main purpose of this paper is to prove that NPRG functions on daf-16. It has been well demonstrated in the daf-16 mutant lifetime experiment. It would be perfect with content determination of phospho-daf-16, which will be supplemented in future studies. |
4 |
In Figure 11, why were standard deviations not shown? How many samples were examined? |
I’m sorry, in the beginning, I plotted with average, but after your reminding, I realized that it was not rigorous, so I plotted with the original data and adopted the analysis of variance (ANOVA) by SAS 8.2. There were three samples for every group. |
5 |
In statistical analysis, t-test is inappropriate for comparing the differences among multiple groups. ANOVA with post hoc tests should be used. Authors should retry their statistical analyses in this study |
In this paper, t-test was widely used to analyze the difference between two groups of data, such as the blank group with NPRR group or the blank group with NPRRP. But for RT‑PCR assay, t-test is inappropriate for comparing the differences among multiple groups, I used ANOVA for analyzing those data. And in statistical analysis, I added that and differences between groups were determined by using analysis of variance. |
Thanks again, best wishes to you!
Round 2
Reviewer 1 Report
The authors have made big improvement on re-organizing figures and correction of mispresentation happened in last manuscript. However, there is a crucial problem still unsolved regarding structural elucidation of the two isolated NPRGs. Structural characterization is the first step through the whole research. As I commented, the information of molecular weight and back bone (sugar chain structure and linkage form) is missing, so there is no way to know the structures of the two polysaccharides only based on monosacchoride composition analysis and even their purity is either unknown. For biological study, if you do not know what exactly you are working with, the result is not persuasive and solid. So I suggest the authors to submit it again after you get the exact structural information and purity of the two compounds.
Author Response
Thank you for your valuable advice!!!
Due to laboratory conditions, we have not been able to determine the back bone of polysaccharides. But we believe that we will use the valuable resources of other laboratories to study these worthy questions in the following research.
Thank you again for your precious energy for this paper!!! Wish you a smooth work and a happy life!
Reviewer 2 Report
In the revised manuscript, authors have addressed most of the reviewer's concerns. However, regarding Comment #2 (the association between NPRR and SOD3 [but not SOD1 and SOD2]), authors' response should be reflected in the revised manuscript.
Author Response
Thanks for your valuable suggestion!!!
I added the following in 2.7 and inserted a document .
Studies have shown that the role of daf-2/daf-16 in regulating the aging and longevity of C. elegans. However, the sod-3 gene that was the downstream factor of daf-16 gene was regulated by daf-2/daf-16 gene, and was related to the life span [18].
Best wishes for you!!!